# Eye Gaze and Self-attention: How Humans and Transformers Attend Words in Sentences

**Joshua Bensemann,** * **Alex Yuxuan Peng, Diana Benavides-Prado, Yang Chen,**
**Neşet Özkan Tan, Paul Michael Corballis, Patricia Riddle,** and **Michael Witbrock**
University of Auckland

## Abstract

Attention describes cognitive processes that are important to many human phenomena including reading. The term is also used to describe the way in which transformer neural networks perform natural language processing. While attention appears to be very different under these two contexts, this paper presents an analysis of the correlations between transformer attention and overt human attention during reading tasks. An extensive analysis of human eye tracking datasets showed that the dwell times of human eye movements were strongly correlated with the attention patterns occurring in the early layers of pre-trained transformers such as BERT. Additionally, the strength of a correlation was not related to the number of parameters within a transformer. This suggests that something about the transformers' architecture determined how closely the two measures were correlated.

## 1 Introduction

Attention is a process that is associated with both reading in humans and with Natural Language Processing (NLP) by state-of-the-art Deep Neural Networks (DNN) (Bahdanau et al., 2015). In both cases, it is the words within a sentence that are attended to during processing. In DNNs, attention results from mechanisms built into the network. Specifically, in the current state-of-the-art method Transformers (Vaswani et al., 2017), this attention process is the result of the dot product of two vectors that represent individual words in the text. For humans, attention processes are more complex as they can be broken into overt and covert attention (Posner, 1980). Overt attention is characterized by observable physical movements of which eye gaze is a well known example that is relevant to reading (Rayner, 2009). Covert attention, on the other hand, is characterized by mental shifts in focus and,

therefore, not directly observable. For this study we have focused on the overt attention measure of eye gaze, with words at the center of an eye fixation being the words that we assume were being attended.

While attention in human reading processes and transformers appear to be completely different, this paper will present an analysis showing the relationship between the two[1]. Specifically, attention in well-known transformers such as BERT (Devlin et al., 2019), and its derivatives are closely related to humans' eye fixations during reading. We observed strong to moderate strength correlations between the dwell times of eyes over words and the self-attention in transformers such as BERT. We have explored some reasons for these different correlation levels and speculated on others.

This analysis is part of an ongoing research line where we attempt to overcome attention limits in transformers. When using transformers, both memory and computational requirements grow quadratically as the sequence length increases because every token attends to all other tokens. In previous work, we have used the attention mechanisms of pre-trained transformers as attention filters that can reduce a sequence length for a sentiment analysis task by 99% while still maintaining 70% accuracy (Tan et al., 2021). Our motivation for this paper was to explore the possibility of using models of eye gaze as an alternative filter. Strong correlations between the attentions produced by transformers and the overt attention of humans would suggest that models of eye movements could potentially be used in computationally inexpensive methods for approximating transformer attention. Alternatively we could use eye movements to train transformer attention towards overt attention patterns[2].

---

Email: josh.bensemann@auckland.ac.nz

[1]Code and Full Results available at `https://github.com/Strong-AI-Lab/Eye-Tracking-Analysis`

[2]See appendix for a preliminary attempt.

## 1.1 Transformers

Transformers (Vaswani et al., 2017) have dominated the leader boards for NLP tasks since their introduction to the deep learning community. Additionally, transformers have had an impact on computer vision (Dosovitskiy et al., 2021), including generative networks (Jiang et al., 2021). The general superior performance of transformers at these tasks is due to its attention mechanism:

$$\text{Attention}(\mathbf{Q}, \mathbf{K}, \mathbf{V}) = \text{softmax}\left(\frac{\mathbf{Q}\mathbf{K}^\top}{\sqrt{n}}\right)\mathbf{V}$$

$$(1)$$

where the word vectors representations of the text sequence $\mathbf{Q}$ are compared to those from sequence $\mathbf{K}$. This is used to determine the amount of information word representations from the former should incorporate from the latter. If the query and key sequence are the same, as in a transformers encoder, it is called self-attention. The results of the attention process are then multiplied by sequence $\mathbf{V}$ to get the final outputs from the attention layer. $\mathbf{V}$ contains different representations for the words in $\mathbf{K}$.

The more relevant a word in $\mathbf{K}$ is to those in $\mathbf{Q}$, the more attention $\mathbf{Q}$ words allocate to that word. Research has examined the $\mathbf{Q}$ x $\mathbf{K}$ part of the attention mechanism to understand how transformers process information. Vaswani et al. (2017) showed that transformers could use words in $\mathbf{Q}$ to learn anaphora resolution by appropriately attending the word "its" in $\mathbf{K}$.

The introduction of transformers was quickly followed by a proliferation of pre-trained models using the transformers architecture. Arguably, the most famous of these models is BERT, a.k.a. the Bidirectional Encoder Representations from Transformers model (Devlin et al., 2019). BERT was designed to encode information from whole passages of text into a single vector representation. Its bidirectional structure means that each word token is placed in the context of the entire sequence instead of just the tokens appearing before it. This structure provided an increase in performance on the GLUE benchmarks (Wang et al., 2019b) over mono-directional models such as the original GPT (Radford et al., 2018).

To ensure that the model learned to attend to the sequence as the whole, BERT was trained using Masked Language Modeling (MLM), a task inspired by the Cloze procedure (Taylor, 1953) from human reading comprehension studies. In MLM, random words from a sequence are hidden during input. The model then has to predict what word was hidden based on the context of surrounding words. BERT was also trained to perform Next Sentence Prediction (NSP) during MLM, forcing words from one sentence to attend to words in other sentences. BERT achieved state-of-the-art performance in multiple NLP benchmarks following this training regime, which led to its widespread adoption.

BERT's impact on the field can be seen in the number of subsequent models that are its direct descendants. Examples include models such as RoBERTa (Liu et al., 2019), which uses BERT's architecture but was trained via different methods. Other models, such as ALBERT (Lan et al., 2020), were created to condense BERT for faster performance with minimal accuracy loss. Even models such as XLNet (Yang et al., 2019) extended BERT's architecture to include recurrence mechanisms introduced in other models (Dai et al., 2019). In turn, some of these descendant models have been used to create other models. For example, BIGBIRD (Zaheer et al., 2020) was built using RoBERTa as its base.

## 1.2 Combining Transformers and Eye Gaze

There is a growing field of research that combines pre-trained transformers with eye-tracking data. Researchers have used outputs from BERT as features for machine learning models to predict eye fixations. In some instances, these outputs are combined with other features (Choudhary et al., 2021); in other instances, BERT itself is fine-tuned to predict eye fixations. For example Hollenstein et al. (2021a) have shown that BERT can be effective at predicting eye movements for texts written in multiple languages, including English, Dutch, German, and Russian.

Given the strong relationship between eye gaze and attention, it is unsurprising that there have been attempts to compare eye gaze to attention generated in transformers. Sood et al. (2020a) compared eye movements in reading comprehension task to three different neural networks, including XLNet. After fine-tuning XLNet, they compared attention from the last encoder layer to eye gaze and reported a non-significant correlation. However, their comparison only reported the correlation for the final attention layer of the network, while other studies comparing transformer attention to human metrics have

indicated that the strength of an association can differ by layer (Toneva and Wehbe, 2019). Therefore, the present study calculated correlations with eye movements from all layers of the transformers. With that said, our results focused on the first layer as it generally produced the strongest correlations to eye gaze data.

Following the work of Sood et al. (2020a), the present study is a large-scale analysis of the relationship between attention in pre-trained transformers and human attention derived from eye gaze. We compared the self-attention values of 31 variants from 11 different transformers, including BERT, its descendants, and a few other state-of-the-art transformers (Table 1). No fine-tuning was performed; models were the same as those reported in their respective papers. Using the BERT-based models with their original parameter weights allowed us to investigate the effect that training regime had on how closely the attention was related to overt eye-based attention. Using non-BERT models allowed us to examine what effect model architecture had on this relationship. Finally, the different datasets enabled an exploration into how the human participants' task also affects this relationship. Results showed significant correlations between attention in the first layer of the transformers and total dwell time. These correlations were unrelated to the size of the model.

## 2 Related Work

There have been attempts to combine DNNs with eye data to perform various tasks. Some basic tasks include predicting how an eye will move across presented stimuli, whether text-based (Sood et al., 2020b) or images in general (Ghariba et al., 2020; Li and Yu, 2016; Harel et al., 2006; Huang et al., 2015; Tavakoli et al., 2017). These predictions can be used to create saliency maps that show what areas of a visual display are attractive to the eye.

In turn, saliency maps can be used to either understand biological visual processes or be incorporated as meta-data into machine learning models. The later endeavor has led to some improvements in task performance. In a recent example, Sood et al. (2020b) achieved state-of-the-art results in a text compression task by creating a Text Saliency Model (TSM) using a BiLSTM network that outputs embeddings into transformer self-attention layers. The TSM was pre-trained on synthetic data simulated by the E-Z reader model (Reichle et al.,

1998) and fine-tuned on human eye-tracking data. The model's output was used to neuromodulate (Vecoven et al., 2020) a task-specific model via multiplicative attention.

Eye gaze data itself can be used to inspire new ways for neural networks to perform NLP tasks (Zheng et al., 2019). For example, it is well known that the human eye does not fixate on every word during reading (Duggan and Payne, 2011). Nevertheless, humans, until recently, performed well above machines in many NLP tasks (Fitzsimmons et al., 2014; He et al., 2021). These observations imply that the word skipping process is not detrimental to reading tasks. Some researchers have exploited this process by explicitly training their models to ignore words (Yu et al., 2017; Seo et al., 2018; Hahn and Keller, 2016). For example, Yu et al. (2017) trained LSTM models to predict the number of words to skip while performing sentiment analysis and found that the model could skip several words at a time and still be as accurate, if not more accurate, than the non-skipping models. Additionally, Hahn and Keller (2018) showed that the skipping processes could be modelled using actual eye movements and achieve the same result. These word skipping models exploit overt attention only, and it would be interesting to know what happens if skipping was modelled on covert attention instead.

Other research exploring the relationship between DNNs and human data has examined how closely the metrics used to measure eye movement are related to metrics used for machine language models. Studies of this type require identifying comparable processes between the two different systems and a suitable dataset. For example, Hao et al. (2020) compared model perplexity to psycholinguistic features.

There have even been comparisons of DNN attention to what humans attend to during reading tasks. Sen et al. (2020) compared the attention of humans during a sentiment analysis task to RNN models. Crowdsourced workers were asked to rate sentiments of YELP reviews and then highlight the important words for their decision-making process. They found correlations between the RNN outputs and human behavior. The strength of these correlations diminished as the length of the text increased.

Closely related to our study is the work of Sood et al. (2020a) who attempted to compare eye gaze

to the attention mechanisms of three different neural network architectures. One of the models was the BERT-based transformer, XLNet (Yang et al., 2019). The other two networks were bespoke CNN and LSTM models. All models were trained on the MovieQA dataset (Tapaswi et al., 2016), and attention values were taken from the later levels of the networks. Several questions for the original dataset were selected for human testing, where the participants' eye gazes were tracked while they read and answered the questions. Sood et al. (2020a) observed that the attention scores from both the CNN and LSTM networks had strong negative correlations with the eye data. However, there was no significant correlation between eye gaze and XLNet.

Finally, there has been recent work using transformer representations to predict brain activity. For example, Toneva and Wehbe (2019) used layer representations of different transformers, including BERT and Transformer-XL, to predict activation in areas of the brain. They found that the middle layers best predict the activation as the context (sequence length) grew. Toneva and Wehbe (2019) tentatively suggested that this means there is a relationship between the layer and the type of processing occurring. To their surprise, they also found that modifying lower levels of BERT to produce uniform attention improved prediction performance.

Schrimpf et al. (2021) performed a similar analysis using many of the models included in the present study. They found that the output of some transformers could be used to predict their participants brain behavior to almost perfect accuracy. Prediction performance differed by model size and training regime, with GPT-2 performing best (Radford et al., 2019). Surprisingly, Schrimpf et al. (2021) found that untrained models also produced above chance prediction, leading them to suggest that the architecture of transformers captures important features of language before training occurs.

## 3 Analysis of Self-Attention Against Eye Gaze

All analyses used HuggingFace's (Wolf et al., 2020) version of the transformer and associated tokenizer. The models' weights were identical to those downloaded from HuggingFace; no fine-tuning was conducted. All analyses report Spearman correlations (Coefficient, 2008) to avoid data normality issues

and provide a direct comparison to previously reported work.

### 3.1 Datasets

Six different datasets were used in our study. In all cases, eye-tracking data were captured from participants performing reading tasks in English.

The GECO Corpus (Cop et al., 2017) contains data from 19 Dutch bilingual and 14 English readers who read "The Mysterious Affair at Styles" by Agatha Christie across four sessions. Comprehension tests occurred between sessions. The bilingual participants completed two sessions in English and two in Dutch. We selected all English sessions for our analysis, regardless of the participant's bilingual status.

The PROVO Corpus (Luke and Christianson, 2018) contains 55 passages (average of 2.5 sentences). Passages were taken from online news articles, magazines, and works of fiction. Participants were 84 native English speakers instructed to read for comprehension.

The ZuCo Corpus (Hollenstein et al., 2018) is a combined reading, eye-tracking, and EEG dataset. Data was captured from 12 native English speakers who could read at their own pace with sentences presented one at a time. The participants completed three different tasks. Task 1 was a sentiment analysis task. Task 2 was a standard reading comprehension task where questions were presented after reading the text. Task 3 was also a reading comprehension task; however, the question appeared onscreen while the participant was reading.

We also used data from Sood et al. (2020a). They collected data from 32 passages taken from the MovieQA (Tapaswi et al., 2016) dataset. In Study 1, 18 participants answered questions from 16 passages under varying conditions such as multi-choice, free answer with text present, and free answer from memory. In Study 2, 4 participants answered multi-choice questions from the remaining 16 passages.

Additionally, we used data from Frank et al. (2013) where 48 participants read 205 sentences from unpublished novels for comprehension. The dataset contains eye movements from both native and non-native English speakers. Participants occasionally answered yes/no questions following a sentence.

The final dataset comes from Mishra et al. (2016) who conducted a sarcasm detection task. The

Table 1: List of models used in this paper

| Model | Pre-trained models in Huggingface repository |
| --- | --- |
| ALBERT (Lan et al., 2020) | albert-base-v1, albert-base-v2, albert-large-v2, albert-xlarge-v2, albert-xxlarge-v2 |
| BART (Lewis et al., 2020) | facebook-bart-base, facebook-bart-large |
| BERT (Devlin et al., 2019) | bert-base-uncased, bert-large-uncased, bert-base-cased, bert-large-cased, bert-base-multilingual-cased |
| BIGBIRD (Zaheer et al., 2020) | google-bigbird-roberta-base, google-bigbird-roberta-large |
| DeBERTa (He et al., 2021) | microsoft-deberta-base, microsoft-deberta-large, microsoft-deberta-xlarge, microsoft-deberta-v2-xlarge, microsoft-deberta-v2-xxlarge |
| DistilBERT (Sanh et al., 2019) | distilbert-base-uncased, distilbert-base-cased, distilbert-base-multilingual-cased |
| Muppet (Aghajanyan et al., 2021) | facebook-muppet-roberta-base, facebook-muppet-roberta-large |
| RoBERTa (Liu et al., 2019) | roberta-base, roberta-large |
| SqueezeBERT (Iandola et al., 2020) | squeezebert-squeezebert-uncased |
| XLM (Conneau et al., 2020) | xlm-roberta-base, xlm-roberta-large |
| XLNet (Yang et al., 2019) | xlnet-base-cased, xlnet-large-cased |

dataset was taken from a wide variety of sources, all short passages containing a maximum of 40 words. Participants were non-native English speakers who were highly proficient in English.

## 3.2 Models

Table 1 lists the 31 variants from the 11 different bidirectional transformers models that we used. Our analysis method required all tokens to attend to all other tokens in a sequence. Therefore, unidirectional models such as GPT-2 (Radford et al., 2019) were excluded as they prevent tokens early in a sequence from attending tokens later in that sequence. We grouped the models into three types: 1) **Basic models** have the same architecture as BERT. 2) **Compact models** are those designed to be smaller versions of basic models. 3) **Alternative models** are those that greatly differ from the basic models.

### 3.2.1 Basic Models

BERT (Devlin et al., 2019): On release, BERT was state-of-the-art. It was trained using MLM, in which 15% of tokens were masked. Training also incorporated NSP by forcing the model to predict whether two sentences were contiguous or not. Our analysis includes a multilingual BERT and both the cased and uncased versions of English BERT.

RoBERTa (Liu et al., 2019): RoBERTa has an architecture identical to BERT but was trained for longer, with larger batch sizes and more data. The MLM examples were dynamically generated during a batch, unlike BERT which used the same mask patterns every time a sample was used. The NSP task was dropped as it did not affect performance.

We have also included the MUPPET version of

RoBERTa (Aghajanyan et al., 2021), trained using multitask learning with tasks from four domains: classification, commonsense reasoning, reading comprehension, and summarization. Finally, we have included XLM-RoBERTa (Conneau et al., 2020), a multilingual version of RoBERTa.

### 3.2.2 Compact Models

ALBERT (Lan et al., 2020): A Lite BERT is a BERT-based model that uses two tricks to reduce the number of parameters and time taken required to train the model. 1) Factorized embedding parameterization - decomposing the large vocabulary embedding matrix into two small matrices; 2) Cross-layer sharing - parameters for all layers are shared.

DistilBERT (Sanh et al., 2019): This model used a Teacher – Student method for the distillation of knowledge (Buciluǎ et al., 2006; Hinton et al., 2015). Sanh et al. (2019) started with a full model and kept every second layer to create the student. The student was then trained on original training data. This procedure resulted in an almost as powerful model but half the size.

SqueezeBERT (Iandola et al., 2020): SqueezeBERT is Bert but with grouped convolutional layers instead of feed-forward layers. The model was trained using the same methods as ALBERT.

### 3.2.3 Alternative Attention Mechanisms

DeBERTa (He et al., 2021): DeBERTa differs from others on this list in that it decouples attention by word semantics from attention by word location. Version 2 of the model used a form of adversarial training to improve model generalization and surpassed human performance on Super GLUE

benchmarks. We have used the RoBERTa based versions in this analysis.

One problem with transformers is the quadratic memory, and computational growth as sequence length increases because every token attends to all other tokens. Some have dealt with this problem by modifying the attention patterns to approximate this full attention pattern without requiring all of the attention comparisons. BIGBIRD (Zaheer et al., 2020) is an example that uses this attention approximation. The model uses a combination of global, sparse, and random attention. Again, we have used the RoBERTa based version of the model.

### 3.2.4 Alternative Architectures

XLNet (Yang et al., 2019): This model is a BERT extension using random permutations of word order during training. The model also incorporates the recurrence mechanism used in Transformer-XL (Dai et al., 2019).

BART (Lewis et al., 2020): BART is an encoder-decoder model that is to recover data from corrupted text input. BART has approximately 10% more parameters than comparable BERT models and no final feed-forward layer. Pre-training was based on corrupting the inputs using token masking, token deletion, token infilling, sentence permutation, and document rotation.

### 3.3 Analysis Method

The transformer data were created by converting the original texts into sentences and then tokenizing those sentences to create sequences. The next step was inputting tokenized sequences into the transformer and extracting the attention matrices produced for each attention head. In terms of Equation 1, we took the output of the softmax function before it was multiplied by $\mathbf{V}$ as that provided a normalized value indicating what proportion of attention each token payed to all others.

The attention value for each token was calculated by averaging across attention heads and matrix rows. This calculation produced a single vector representing the amount of attention allocated to each token by all others in the sentence. Our procedure differs from Sood et al. (2020a) who used the maximum attention from each word instead of the mean. Some preliminary analyses suggested that the mean attention values provided more stable results across datasets. The results using the maximum values are available on our GitHub repository for comparison purposes. If a word was tokenized

into sub-words, those sub-words were also averaged to produce a single value. The special tokens [CLS] and [SEP] were used for the attention calculations but dropped from the final word-level attention vector. Finally, attention was normalized by sentence by calculating the proportion of attention allocated to each word.

Dwell time was used for the overt human attention data. Dwell time is a measurement of the total time that a participant's eye fixated on a word. This choice was necessary for consistency between analyses as it was the only measure to appear in all datasets. Dwell time data was extracted for each word in a sentence, with one sentence being produced for each participant in the original data. The dwell time data were also normalized by sentence by calculating the dwell time proportion for each word. The data from individual participants were then averaged to create one normalized sentence for each sentence in the text.

Data from the transformers and human participants were then matched so that each word in the text had a sentence normalized attention score from the transformers and the average participant. After matching, all the words from a text were pooled and used to calculate the Spearman correlation values. One-word sentences were removed as both scores were always 1.0, which inflated the correlation scores.

### 3.4 Results and Discussion

There were significant positive correlations between the total dwell time and the attention from all layers of the different models. This finding was an apparent departure from the results of Sood et al. (2020a) who reported a non-significant correlation of -.16 between the last layer of XLNet and their dataset. For comparison, we obtained a .428 correlation for their Study 1 data and .327 for their Study 2 data from XLNet's last layer. Although they did not directly specify the normalization they used, we suspect that the difference in results is due to us using sentence-level normalization and Sood et al. (2020a) using paragraph normalization. For comparison, we ran the same procedure using paragraph normalization and obtained non-significant correlations just as they did. In general, many of the correlations obtained using sentence normalization become much weaker when using the paragraph normalization. This finding corresponds well with the Sen et al. (2020) finding that attention for

## Highest Correlation

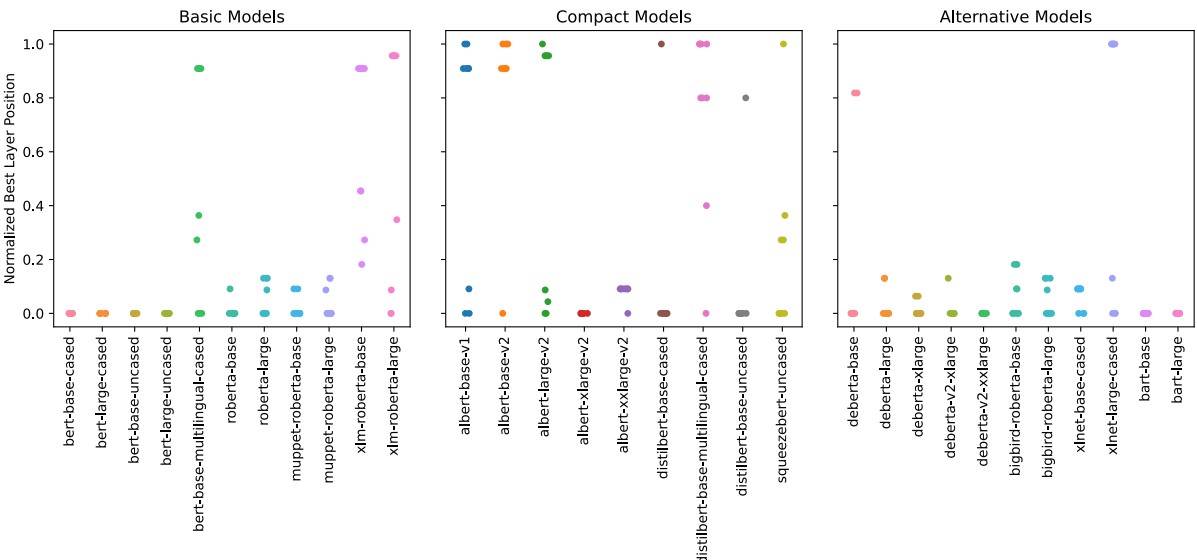

Figure 1: The relative position of the layer with the highest correlation. 0 is the first layer, 1 is the last layer. There are multiple dots for each model because each dot represents the highest correlation from a different dataset.

## 1st Layer Correlations

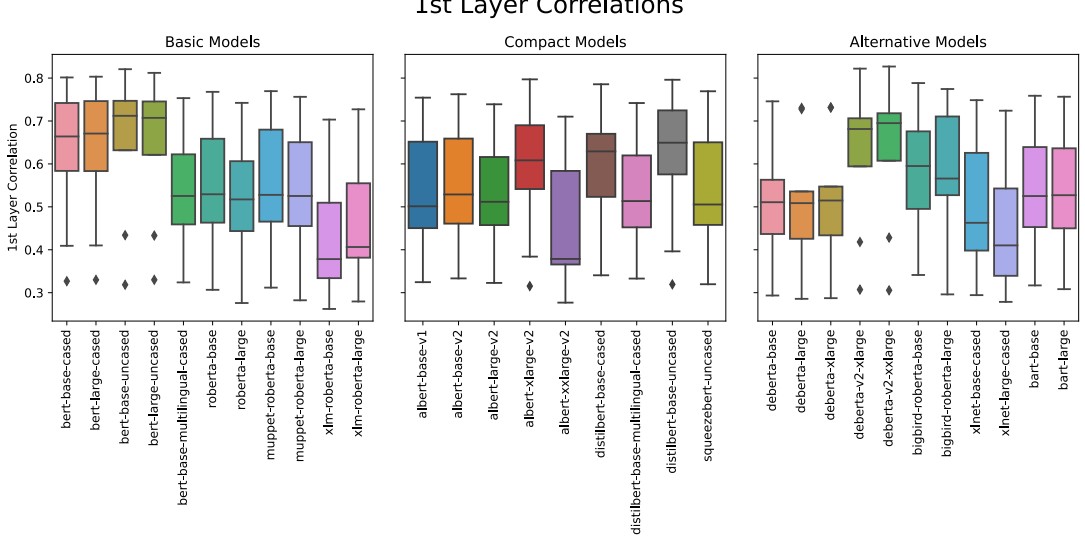

Figure 2: The correlations between the first layer attention patterns and eye gaze data from all models. The box plots represent the spread of correlation values across datasets.

non-transformer neural networks became less correlated with eye movements as the length of the text increased. All analyses presented here refer to sentence-level correlations. Paragraph-level analyses can be found in our GitHub repository.

Our first analysis investigated which attention layer was most closely correlated with the eye gaze data. Figure 1 shows the relative position of the layer with the highest correlation by model. In many cases, the highest correlation was produced by the earlier layers of each model, in 66.2% of

cases this was the first layer (position 0). Notable exceptions to this rule are the multilingual versions of BERT and RoBERTa (i.e., XLM) and many compact models. Although further studies are needed, the finding that multilingual variants of models do not behave like monolingual variants is in line with some previously reported studies (Conneau et al., 2020; Hollenstein et al., 2021b; Vulić et al., 2020), where some studies report multilingual benefits and while others do not.

Further investigations found that when the first

Table 2: First layer correlations By dataset. Strongest correlations have been bolded.

| Model | GECO | Mishra | Provo | Sood S1 | Sood S2 | ZuCo S1 | ZuCo S2 | ZuCo S3 | Frank et al |
|---|---|---|---|---|---|---|---|---|---|
| albert-v1 | 0.744 | 0.754 | 0.497 | 0.450 | 0.326 | 0.501 | 0.580 | 0.325 | 0.652 |
| albert-v2 | 0.748 | 0.739 | 0.492 | 0.460 | 0.329 | 0.503 | 0.585 | 0.326 | 0.637 |
| bart | 0.729 | 0.758 | 0.526 | 0.451 | 0.323 | 0.511 | 0.550 | 0.313 | 0.638 |
| bert-cased | 0.802 | 0.783 | 0.668 | 0.584 | 0.410 | 0.643 | 0.679 | 0.328 | 0.744 |
| bert-multilingual-cased | 0.753 | 0.727 | 0.525 | 0.459 | 0.338 | 0.489 | 0.622 | 0.324 | 0.603 |
| bert-uncased | 0.816 | **0.791** | **0.710** | **0.626** | **0.434** | **0.693** | **0.722** | 0.324 | **0.746** |
| birdbird-roberta | 0.775 | 0.774 | 0.600 | 0.511 | 0.363 | 0.582 | 0.565 | 0.319 | 0.693 |
| deberta-v1 | 0.731 | 0.735 | 0.511 | 0.432 | 0.310 | 0.502 | 0.533 | 0.289 | 0.549 |
| deberta-v2 | **0.824** | 0.770 | 0.708 | 0.601 | 0.423 | 0.688 | 0.712 | 0.306 | 0.660 |
| distilbert-cased | 0.786 | 0.772 | 0.623 | 0.523 | 0.378 | 0.629 | 0.632 | 0.341 | 0.670 |
| distilbert-multilingual-cased | 0.742 | 0.740 | 0.513 | 0.452 | 0.337 | 0.487 | 0.620 | 0.333 | 0.602 |
| distilbert-uncased | 0.796 | 0.780 | 0.649 | 0.576 | 0.396 | 0.649 | 0.678 | 0.319 | 0.725 |
| roberta | 0.709 | 0.755 | 0.523 | 0.453 | 0.329 | 0.504 | 0.537 | 0.291 | 0.632 |
| roberta-muppet | 0.712 | 0.763 | 0.527 | 0.460 | 0.329 | 0.501 | 0.542 | 0.297 | 0.665 |
| squeezebert | 0.730 | 0.769 | 0.505 | 0.458 | 0.320 | 0.499 | 0.549 | **0.348** | 0.650 |
| xlm | 0.690 | 0.715 | 0.391 | 0.358 | 0.271 | 0.379 | 0.476 | 0.313 | 0.532 |
| xlnet | 0.678 | 0.736 | 0.436 | 0.369 | 0.287 | 0.408 | 0.470 | 0.297 | 0.584 |

layer did not produce the highest correlation, the first-layer correlation value was on average, 0.055 lower than the best correlation value. In 75% of cases, this difference was less than 0.082. Therefore, the first layer value appears to be a good representation of the correlation between the model and the eye gaze data. An extreme example of this were the ALBERT variants, which, likely due to weight sharing during training, have virtually identical correlations from attention values from each of its levels (Figure 3). Due to its general best performance, the first layer results have been used at the best performance for all models. Analyses using the actual best performance can be observed in our GitHub repository, although those results are highly similar to those reported here.

Our next analysis compared performance across models based on the first layer correlations. Figure 2 shows that, in general, the size of the model does not determine the correlation between the human eye and transformer attention. Evidence for this can be seen in minor differences between various-sized variants of the same model. For example, the cased and uncased versions of BERT-base and BERT-large are very similar, despite the large variant containing 340 million parameters compared to the base variants' 110 million. Similar observations can be observed across the other models, especially DeBERTa, where the largest variants have 1.5 billion parameters, and the smaller ones contain less than 1/3 of that number. This observation was confirmed with a non-significant sign test ($p = .090$) that compared each variant to the next smallest variant in its model type. Due to this simi-

larity, results in Table 2 reports a single value per model type that is an average for each size variant. Table 3 shows the highest correlation by dataset. In most cases, this model was either BERT-uncased or DeBERTa-V2.

While the number of parameters is not what determines the correlations, comparing across models in Figure 2 suggests that training is essential for determining those relationships. For example, the BERT models have identical architectures to various RoBERTa models, yet Table 2 shows that the BERT correlations were consistently higher than the RoBERTa based models. The other clear examples of training effects can be seen in the differences between DeBERTa V1 and V2, where V2 models use the Scale-invariant-Fine-Tuning (SiFT) algorithm introduced in the original paper. Interestingly, the addition of the SiFT algorithm allowed DeBERTa V2 to surpass human performance on the SuperGLUE benchmarks (Wang et al., 2019a), and Table 3 shows that this model was often the second-highest correlated model. While it would be great to find a direct relationship between how human-like a model's performance is and how correlated its attention patterns are to eye movements, that is not the case. Excluding the compact models, the BERT descendants outperform it on many of the benchmarks, yet only DeBERTA comes close to having stronger correlations to human eye movements. In most cases, attention patterns less correlated with overt human attention produced better overall performance on NLP tasks.

Tables 2 and 3 show the rankings by correlation are similar between datasets, with BERT-uncased

Table 3: The three models with strongest correlation to eye-tracking data for each dataset. The uncased version of BERT produced the strongest correlation in 7 out of 9 cases.

|   | GECO | Mishra | Provo | Sood S1 | Sood S2 | ZuCo S1 | ZuCo S2 | ZuCo S3 | Frank-et-al |
|---|------|--------|-------|---------|---------|---------|---------|---------|-------------|
| 1 | deberta-v2 | bert-uncased | bert-uncased | bert-uncased | bert-uncased | bert-uncased | bert-uncased | squeezebert | bert-uncased |
| 2 | bert-uncased | bert-cased | deberta-v2 | deberta-v2 | deberta-v2 | deberta-v2 | deberta-v2 | distilbert-cased | bert-cased |
| 3 | bert-cased | distilbert-uncased | bert-cased | bert-cased | bert-cased | distilbert-uncased | bert-cased | distilbert-multilingual | distilbert-uncased |

producing the highest correlation in all but two cases. In one of the exceptions, the GECO dataset, BERT-uncased, was ranked second. In the other exception, ZuC0 Task 3, the ranking was much lower. In general, the correlations from ZuCo Task 3 differ greatly from the other datasets. The correlations are lower for all models, and the model rankings are very different, with two of the compact models, SqueezeBERT and DistillBERT, ranking highest, and BERT-uncased, ninth. Task 3's participants were the same as Tasks 1 and 2. Those first two tasks produced results closer to the other datasets, meaning Task 3's lower correlations were likely due to the task itself.

Interestingly, in Task 3, the participants were presented with the question on the screen, allowing them to direct their eye gaze to find the information they required. This contrasts with most of the other datasets where the questions about the data were presented after reading. The only exceptions to this were some tasks by Sood et al. (2020a) where the question appeared on screen in Study 2 and in 2/3s of the tests in Study 1. Furthermore, the correlations from Sood et al. (2020a) Studies 2 and 1 were also the second and third lowest of the datasets, respectively (Table 2). While further study is needed, the lower correlations from SOOD et al. and ZuCo Task 3 may indicate that while transformer attention patterns produce strong correlations when reading typically, the relationship drops when the reader actively searches for information.

Our final analysis looked at correlations across levels of BERT (Figure 3). The results of Toneva and Wehbe (2019) suggest that the middle layers of BERT provided the best features for predicting brain activity in humans. They speculated that these relationships could mean that the middle layers of BERT could be related to the kinds of processing that occurs in those brain levels. Our results show that the attention patterns from BERT's first layer were closely related to eye gaze data. Again, while speculative, our results combined with Toneva and Wehbe (2019) would suggest that for BERT at least, the lower levels correspond best

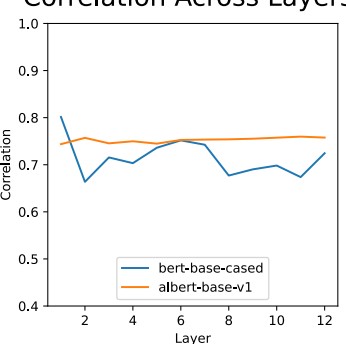

Figure 3: The average correlations across layers for bert-base-cased and albert-base-v1.

to text information entering the eyes. In contrast, the middle layers correspond to specific processing. With that said, not all transformers produced the strongest correlations from their first layer. For example, as mentioned above, Figure 3 shows the data from ALBERT-V1 where the correlations from all levels were relatively the same.

## 4  Conclusion

This paper analyzed the correlations between attention in pre-trained transformers and human attention derived from eye gaze. We found correlations between the two that were generally stronger in the earlier layers of the model and, in most cases, strongest in the first layer. These correlations were unaffected by the model's size, as different sized variants of models produced similar correlations. The training the models received did appear to matter, although the present study cannot determine the full extent of that relationship. We found that correlations were weaker from eye-tracking studies where the participants could actively guide their reading towards seeking the information they needed than when presented with questions after reading. While we found a relationship between overt human attention and attention in some pre-trained transformers, additional research would be required before models of eye gaze could be used to replace attention in transformers.

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

## A Investigating the Effect of Injecting Eye Gaze Bias During Training

As a preliminary experiment, we investigated the effect of injecting human eye gaze bias during training on test accuracy. We used the BERT model (Devlin et al., 2019) and the sarcasm-detection dataset published in Mishra et al. (2016) as a case study.

### A.1 Method

The Mishra et al. (2016) dataset was originally proposed to predict non-native English speakers' understanding of sarcasm by using eye-tracking information. The dataset contains information on the fixation duration of each word for each participant. We injected the eye-gazing bias during training by optimising the following loss function:

$$L = H(y, \hat{y}) + \alpha H(p, \hat{p}) \qquad (2)$$

where $H(y, \hat{y})$ is the cross-entropy loss of the binary classification task of sarcasm detection, and $H(p, \hat{p})$ computes the divergence of the first-layer attention values from the distribution of the normalised fixation duration values given a sentence. The hyperparameter $\alpha$ controls the weight of the second term in the loss function.

Our experiments only used the fixation duration values from Participant 6 because they had the highest overall accuracy for sarcasm detection (90.29%). All the hyperparameters were tuned on a validation set extracted from the training set before being applied to the entire training set.

### A.2 Results

The results are plotted in Figure 4. As expected, the models fine-tuned from pre-trained BERT models had significantly better test accuracy on both the small and large training sets than models trained from scratch on the Mishra et al. (2016) dataset.

A t-test confirmed that when the models were trained on the large training set without pre-training, an eye gaze bias injection during training hurt the performance ($p < .05$). With pre-training, both models in Figure 4(b) performed better than the best participant in the Mishra et al. (2016) dataset. The bias injection still lowered the mean accuracy, although the difference was no longer statistically significant. When the small training set was used to train the models, we found no significant difference after the bias injection.

Comparing our results to Sood et al. (2020b) suggests that training a model to predict eye gaze

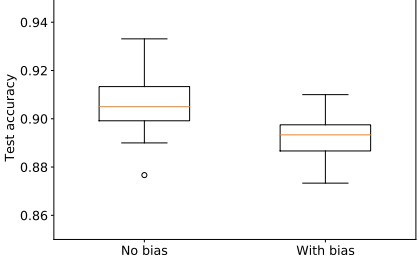

(a) Large training set without pre-training

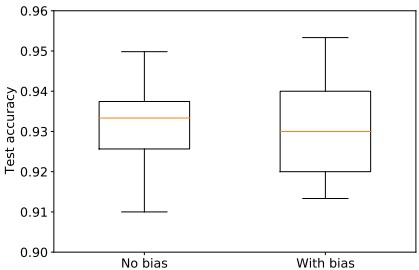

(b) Large training set with pre-training

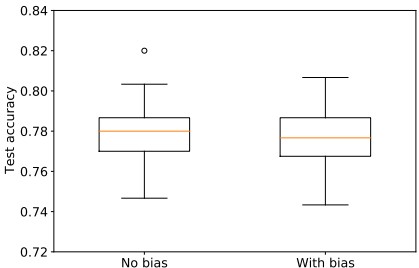

(c) Small training set without pre-training

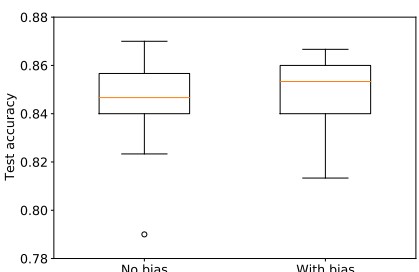

(d) Small training set with pre-training

Figure 4: Comparison of the BERT models trained with eye gaze bias against the models trained without in terms of test accuracy. Models in plots (a) and (b) were trained on 693 examples, and the results were obtained after 20 runs. Models in plots (c) and (d) were trained on only 70 examples, and the experiments were repeated 50 times. The same test set (300 examples) was used for all the experiments.

improves text compression performance, whereas using eye gaze data to regulate sarcasm detection decreased performance. It is unknown whether the difference in results is due to our task choice or to our method of using human data.