# OpenReview forum: "Eye Gaze and Self-attention: How Humans and Transformers Attend Words in Sentences"
_aclweb.org/ACL/2022/Workshop/CMCL — CMCL 2022_

### Official Review · Reviewer_oK4s · 2022-03-21
**Human vs machine attention during reading**

**Rating:** 7
**Confidence:** 3

**Review:**

 ## Overall

This paper analyzes the degree of correspondence between human eye fixation patterns during text reading and the attention distributions of pretrained transformer language models. Results show an overall high degree of correlation between model-generated attention weights and human gaze distributions across transformer architectures and layers, suggesting that transformers may converge via pretraining on a relatively human-like assessment of the importance of words in texts. Although I have some concerns about framing and methods, I think this paper should be of interest to computational psycholinguistics.


## Major

- Framing and motivation leave something to be desired. First, attention in the deep learning toolkit and attention as a cognitive process happen to be referred to with the same word in English, but that doesn't entail that the relationship between them is inherently scientifically interesting. The authors' assumed stance on the relationship between these two constructs is confusingly presented: the first sentence of the paper implies that both attentions are considered to be the same thing, whereas the first sentence of the 2nd par implies that they are "appear to be completely different". In reality, "attention" in DNNs is a cognitively inspired architectural innovation that improves task performance but whose role in the model's reasoning process is not well understood (e.g. Jain & Wallace 19, Serrano & Smith 19), let alone its relationship to human cognition. The paper would benefit from a clearer framing of the constructs under investigation. Second, why does the relationship between model and human attention matter? What do we learn about models or humans from a particular level of correlation between attention weights and eye gaze? To be clear, I'm *not* saying that these questions don't matter, only that it's on the authors to explain what's at stake. The paper dives right into the details without clearly setting up the questions that the authors want to answer.

- Human gaze distributions were averaged prior to analyses. Because reading patterns are known to be highly variable across individuals, I think this is a serious weakness. In particular, it may lead to an exaggerated impression of human-model similarity by eliminating a major source of variation in the eye movement record. More realistic estimates could be derived from e.g. regression models of detailed (unaveraged) fixation distributions given model attention proportions, ideally in a mixed effects design. In addition, interpretation is made difficult by the absence of baselines. How good is 0.6 Spearman correlation, and what drives the correlations? Without controls, it's hard to say whether this is really about the model's learned attention vs less interesting phenomena that attention may merely reflect in some way, like the length or frequency of words.

- This paper surveys a lot of the models in the huggingface inventory but the absence of GPT (and variants) is a liability, especially since GPT is one of the only SOTA transformer LMs that respects incrementality (a constraint faced by human readers) and produces representations that are substantially more similar to the human brain response to language than many of the other models considered here (Schrimpf et al 21, cited here). GPT is available from the same software library as the others -- why was it omitted?

- Several critical claims are made based on numerical differences without direct statistical comparison. For example, none of the following claims are directly supported quantitatively:
    - Lower layers are better correlated with human gaze than higher layers
    - There is no relationship between the size of the model and attention-gaze correlation (this one actually isn't quantified at all, e.g. via correlation between nparams and spearman corr)
    - BERT is better overall than the other models
    - Attention-gaze correlations are higher during free (vs task-directed) reading
    - Lower layers model eye movement control whereas middle layers model specific language processing operations


## Minor

- Rai & Le Callet 2018 is a suboptimal single citation to provide for the overt-covert attention distinction. Posner 1980 would be better.
- Fig 1 is hard to interpret. Sure, many maxima are in early layers, but there are plenty of exceptions, and many from the highest layers. What's the pattern? How different are the layers from each other? A distribution would be a lot more informative here.
- Ettinger 20 is cited as a study about eye movements vs machines, but they didn't look at eye movements. This cite should either be removed, or the context should be rephrased.
- This paper is heavily inspired by Sood et al 20 and cites it frequently, but the presentation of that earlier work is strange. In particular, it's never said clearly enough that this paper is asking a different question from Sood et al 20. Sood et al 20 asked whether *model performance* on a QA task was anticorrelated with the *similarity* between model attention distributions and human gaze distributions. They showed that models that do well on the QA task have more similar attention distributions to humans, which is not the same as this paper's question about the *overall similarity* of human and model attention, independent of task. This is why Sood et al's associations are negative, which is an otherwise confusing outcome as presented here. It's also not clear why the authors feel beholden to account for "departures" of their findings from Sood et al's results at the start of S3.3. I'm glad this resulted in some interesting error analysis about the importance of normalizing over sentences vs paragraphs (though see next point), but I don't understand the reason for the question in the first place -- Sood et al and this paper are analyzing different things in response to different questions, so why would the results be expected to be the same?
- The finding that model-human correlations are higher when normalizing over sentences vs paragraphs is presented as a puzzle and the authors speculate that model-human alignment may degrade with longer texts. However, the texts and model attention distributions are the same in both analyses (only the domain of normalization changes), so this explanation doesn't make sense to me. Plus isn't matching at the level of paragraphs strictly harder than matching at the level of sentences, since at the paragraph level, the model needs to reproduce the gaze distribution of each component sentence (i.e. the sentence-level task) *and* the overall proportion of gaze across sentences? So I think the degradation when normalizing over paragraphs just falls out from sentences being substrings of paragraphs. Which domain of normalization is more appropriate depends on the scientific question, which isn't fully spelled out here (see major point 1).
- The paper puts all description of datasets in the appendix, but some high level description should be provided in the main text, especially since they go on to analyze differences between datasets/tasks in S3.3.
- There are known potential problems with averaging bounded variables like normalized attention distributions: because the bound isn't taken into account in the average, values in the center of the range can have an excessive influence, and the degree of concentration toward the extremes can be poorly estimated. In the domain of Pearson correlation, this is often handled with Fisher transformation. Have the authors considered and tried to address possible influences of edge effects on their results?
- To compute model attention distributions, averages were first taken across attention heads and token-level attention distributions, and then the weights on subword tokens were averaged to bring the tokenization into alignment with reading experiments. But it seems like a more principled method would first *sum* the attention weights allocated to each subword before any other averaging, since this sum represents the total proportion of attention allocated to the entire word.

---

### Official Review · Reviewer_YKVU · 2022-03-24
**Review of eye-gaze and self-attention**

**Rating:** 5
**Confidence:** 3

**Review:**


The paper discusses an approach to modeling eye-gaze using BERT and similar models. In particular it shows that dwell time in human eye-tracking data correlated with attention patterns in early layers of these models. Additionally, this correlation was found to be weak on data where participants had to respond to questions after reading sentences.

I thought it was very well written paper. Tha authors do a wonderful job in stating the assumptions, in describing the hypothesis, situating their work in the previous literature as well as in discussing the results.

My key concern is with regard to the interpretation of the correlation and indeed, with the hypothesis of relating attention in transformers with human overt attention (operationalized via gaze duration/dwell time). This is because of the underlying nature of the model like BERT -- the authors state "Its bidirectional structure means that each word token is placed in the context of the entire sequence instead of just the tokens appearing before it." Now, it seems to me that while such a mechanism could be used to model grammatical knowledge but it's difficult for me to understand how it could model an online process such as overt attention. This is because when English native speakers read a word in a sentence they have very limited access to the context on the right of a word that they are fixating. It is true that the authors are modeling gaze duration (rather than a measure like first fixation duration) which implies that the measure captures reading time due to regressions as well (i.e., looking at the word again after accessing right context), but we know that regressions are rare during reading. So, regressions being rare while typical reading implies that reading doesn't involve use of right context. Can this be said of the models being use to model overt attention? Indeed, given the nature of how BERT works one would expect the correlations to be in fact stronger in cases where reading involves active search for information (i.e., when reading to answer questions). This is because one could expect more rereading in such a setting.

So, my main concern is that the 'overt attention' - 'transformer attention' link that has been established in this paper is rather tenuous and in my understanding theoretically weak. I would request the authors to at least acknowledge this limitation of the work in the final draft, if the paper were to get accepted. The authors state that "... this paper will present experimental evidence of a link showing the relationship between the two." I suggest that the author qualify the nature of this link; as it is currently stated, the use of the word 'link'  implies process parallelism which as discussed above is very misleading.

---

### Official Review · Reviewer_zT3s · 2022-03-27
**Attention in humans and neural systems**

**Rating:** 7
**Confidence:** 3

**Review:**

This paper investigates the relationship between human attention (in the form of eye movements) and the attention used by pre-trained transformers. The authors perform a nice comparison between the way humans and multiple models process text. All in all, I find this paper well written and definitely of some interest to the CMCL community. For this reason, I would be happy to see it published.

Said so, I would still like to read more about the following aspects. The authors could easily shrink the related work section a little bit and gain the half-a-column necessary for clarification.

* You claim that the first/early layers show the highest correlation overall. It is important to support this claim with some analysis and a bit more discussion on why we are seeing this pattern and what we can conclude.

* Directionality is always tricky when dealing with human processing. Why not addressing it? There are plenty of uni-directional models that you could include in the analysis as a comparison point.

* Given the audience, it would be nice to read more about the reason why it is important to draw a line between attention in humans vs. models. You could specify if such correlation can tell us more about processing in general.

---

### Decision · Program_Chairs · 2022-03-29

Accept